# The relationship between finger length ratio, masculinity, and sexual orientation in women: A correlational study

**Luke Holmes**[1]*, **Tuesday M. Watts-Overall**[2], **Erlend Slettevold**[3], **Dragos C. Gruia**[4], **Gerulf Rieger**[1]

1 Department of Psychology, University of Essex, Colchester, United Kingdom, 2 School of Psychology, University of East London, Stratford, United Kingdom, 3 Faculty of Medicine and Health Services, University of East Anglia, Norwich, United Kingdom, 4 Department of Brain Sciences, Imperial College London, South Kensington, United Kingdom

* lukeholmesphd@gmail.com

## Abstract

Homosexual women are, on average, more masculine in their appearance and behavior than heterosexual women. We hypothesized that their masculinity was influenced by exposure to elevated levels of prenatal androgen during early development. We recruited 199 women (including 67 homosexual women) and measured their masculinity via self-report and observer ratings. Our measure of prenatal androgen exposure was the ratio of the index to ring finger (2D:4D), which is hypothesized to be lower in women exposed to elevated levels of androgens during prenatal development. Homosexual women were substantially more masculine than heterosexual women in both self-report and observer ratings. However, homosexual women neither had more male-typical finger length ratios, nor did their finger length ratios relate to their masculinity in any predicted direction. Thus, we found no evidence that increased prenatal androgen exposure influenced masculinity in homosexual women.

## Introduction

Research on masculinity and femininity shows a consistent pattern with respect to women's sexual orientation: Homosexual women recall or report more masculine and less feminine behaviors and self-concepts in childhood and adulthood [1, 2], and report more male-typical and less female-typical interests than heterosexual women in adulthood [3–5]. Longitudinal work also suggests that early childhood masculinity is robustly associated with homosexual attractions in females later in life [6, 7]. Furthermore, based on evaluations by others of their childhood photographs and videos, girls who identified as homosexual in adulthood were rated as more masculine and less feminine than girls who later identified as heterosexual [8, 9]. In adulthood, homosexual women are also perceived as more masculine than heterosexual women [2, 10]. These perceptions by others are particularly valuable, because self-reports of masculinity-femininity are possibly biased due to social desirability [11]. Thus, in the present study, we included evaluations of participants by observers, in addition to self-reports, to verify the link of female sexual orientation with masculinity-femininity with multiple measures.

**Data Availability Statement:** The data is being held in the UK Data Service, and is available here: https://reshare.ukdataservice.ac.uk/854580/ The doi is set to go live in 24-48h at the time of writing,

but can be found here when it is up: 10.5255/
UKDA-SN-854580.

**Funding:** This research was supported by grants
awarded to G. R. by the American Institute of
Bisexuality (RSC2519; https://www.
americaninstituteofbisexuality.org/) and the
University of Essex (DG00832; https://www.essex.
ac.uk/). The funders had no role in study design,
data collection and analysis, decision to publish, or
preparation of the manuscript.

**Competing interests:** The authors have declared
that no competing interests exist.

Androgen exposure during prenatal development is a potential explanation for the link between sexual orientation and masculinity-femininity within each sex, in addition to explaining overall differences in masculinity-femininity between males and females [12, 13]. For example, females with Congenital Adrenal Hyperplasia (CAH), which results in elevated androgen exposure in early gestation, are more likely than their unaffected siblings to engage in male-typical behavior during childhood [14], and report same-sex sexual attractions during adulthood [15].

However, most research on the subject of prenatal androgen influences in humans is informed by postnatal measures, which are assumed to reflect early exposure. The most prominent of these is the ratio of the length of the second to fourth finger digits (2D:4D). Men have lower 2D:4D than women [16], and this sex difference emerges in early gestation [17]. Moreover, women with CAH also have lower (more male-typical) 2D:4D than other women, possibly due to the increased androgens exposure [18]. Homosexual women have more male-typical 2D:4D than heterosexual women, on average [19, 20]. This effect was confirmed in both their left and right hands in a meta-analysis, Hedge's $g's$ = .23 and .29, .04 < 95% CIs < .51. In contrast, homosexual men do not robustly differ in 2D:4D ratio compared to heterosexual men [16].

It should be noted that 2D:4D as a measure of prenatal androgen exposure remains a controversial topic because of ongoing debates about causation [21, 22], validity due to small effects in the presence of noise in the data [23], and the possibility of publication bias in the literature [16]. That is, the meta-analysis by Grimbos et al. (2010) estimated a given amount of unpublished data with null findings. Once included in their main analysis, these estimated null findings reduced the link between women's sexual orientation and their digit ratios from Hedge's $g's$ = .23 and .29, for the left and right hand, to .07 and .13, respectively. Thus, even though reasonable arguments can be made that 2D:4D reflects early androgen exposure related to sexual orientation [12], the exact strength of the relationships between these traits remains unclear. For this reason, studies using 2D:4D as a measure as prenatal androgen exposure must be interpreted with caution.

In sum, there may be a relationship between female sexual orientation and masculinity-femininity, and between female sexual orientation and 2D:4D. It is possible that these patterns are further associated. For example, prenatal androgen exposure, possibly reflected by 2D:4D, could be the common factor that influences both women's sexual orientation and their degree of masculinity-femininity. If this is the case, one could expect that homosexual women's increased masculinity, in comparison with heterosexual women, is explained by the finding that homosexual women have, on average, more male-typical 2D:4D than heterosexual women. Hence, the differences in 2D:4D across all women could mediate the relationship between their sexual orientation and their degree of masculinity.

Another possibility is that an interaction between sexual orientation and 2D:4D explains why certain women show a greater degree of masculinity in their behaviors and self-concepts. There is significantly more variability in measures of masculinity-femininity among homosexual women than heterosexual women, because some homosexual women are especially masculine, compared both with heterosexual women and other homosexual women. Homosexual women's greater degree of variability in their masculinity has been repeatedly reported in different studies, and with different measures of masculinity-femininity, including both self-reports and observer ratings [2–4, 9, 24, 25]. For instance, Lippa's (2005) meta-analysis showed that homosexual women were more variable in self-reported masculinity-femininity than heterosexual women, with a mean variance ratio of .67 between the groups. In other words, some homosexual women are especially masculine compared with both heterosexual women and other homosexual women. It is possible that the most masculine homosexual women, in

particular, have been exposed to elevated levels of androgens during early development. Hence, homosexual women who have the most male-typical markers of androgen exposure, such as the most male-typical 2D:4D, may also be the women that are the most masculine, as compared to both heterosexual women and other homosexual women with less male-typical 2D:4D. This line of reasoning points to a potential interaction between 2D:4D and sexual orientation, predicting degree of masculinity.

One previous study provided support for the hypothesis that variability in homosexual women's self-reported masculinity-femininity is partly explained by their differing degrees of androgen exposure. Homosexual women who self-identified as "butch" (i.e., masculine) had significantly more male-typical 2D:4D than those who self-identified as "femme" [i.e., feminine; 26]. This finding suggests that there may be different types of homosexual women, with prenatal androgen exposure possibly being the developmental factor which distinguishes between them. A related study treated butch-femme as a continuous variable, and found for the left hand (but not the right hand) that "more butch" participants had lower (i.e. more masculine) finger length ratios than "more femme" participants [27]. Another study measured homosexual women's reported roles during sex (classed as "butch/active" versus "femme/passive") and found no association between reported sex roles and their level of 2D:4D [19]. However, because sex roles of homosexual women may simply not equate to their degree of masculinity-femininity [28], we considered the findings by Brown et al. (2002) as potentially more informative with respect to the hypothesis that variation in homosexual women's masculinity-femininity is explained by differences in their 2D:4D.

If it is the case that there is more variability in masculinity-femininity in homosexual women than in heterosexual women, and this is explained by differences in their digit ratios, then it could also imply that there is more variability in 2D:4D among homosexual women than heterosexual women. To our knowledge, no previous research has examined the degree of variability in 2D:4D across women with different sexual orientations. The present research examined this pattern.

Our discussion thus far has focused on a comparison of heterosexual and homosexual women. Bisexual women appear to be intermediate between heterosexual and homosexual women in their masculinity-femininity [3, 4]. We are not aware of research that specifically compared 2D:4D of bisexual women to those of other women. Bisexual women were included in the present research. However, in order to ease interpretation, we mostly focus on comparisons between heterosexual and homosexual women, and bisexual women are revisited in the Discussion.

Based on the reviewed literature, the following hypotheses were tested with the measure of 2D:4D as indicator of prenatal androgen exposure, in addition to three measures of masculinity-femininity taken during this research: self-recall from childhood, self-report from adulthood, and observer ratings of adulthood behaviors:

Hypothesis 1. Homosexual women are, on average, more masculine than heterosexual women by both self-report and via observer ratings.

Hypothesis 2. Homosexual women have, on average, more male-typical (lower) 2D:4D than heterosexual women.

Hypothesis 3. The relationship between sexual orientation and masculinity in women is mediated by their male-typical 2D:4D.

Hypothesis 4. Homosexual women are, on average, more variable than heterosexual women in both their masculinity-femininity and their 2D:4D.

Hypothesis 5. Homosexual women with the most male-typical 2D:4D show the greatest degree of masculinity, as compared to heterosexual women and other homosexual women with less male-typical 2D:4D.

## Method

### Participants

**Target participants.** In planning our sample size, we drew upon previous studies which used identical measures, and which had computed the correlations of sexual orientation with either masculinity-femininity, or with 2D:4D. Correlations ranged from .30 (for 2D:4D) to .40 to .60 [for measures of masculinity-femininity; 2, 20]. A power analysis conducted in G*Power determined that a minimum of 112 women would be necessary for the smallest estimated main effect ($r = .30$) to achieve significant results with a power of .90.

With regards to the moderation and mediation, estimating the necessary sample size proved difficult, as no other study has conducted a moderation or mediation in the same manner as the present study. As such, we erred on the side of caution with participant numbers: Our power analysis for the main effect was based on the more conservative power value of .90 rather than the commonly-used .80, resulting in a sample size requirement of 112 instead of 82 for the smallest expected main effect ($r = .30$). Additionally, we continued recruiting past this figure as participants were visiting our laboratory for other studies regardless, resulting in a final sample size of 199 –almost double the requirement for the estimated main effects to reach significance. However, even though we did everything we could to ensure a lab-based study such as ours was sufficiently powered, the uncertainty regarding power of the moderation and mediation analyses in particular should be noted, and we revisit this limitation in the Discussion.

A total of 199 women were recruited from Colchester and London, United Kingdom via Pride festivals, online news sites for homosexual women, and university mailing lists. Using a 7-point sexual orientation scale [29], women self-identified as "exclusively straight" (n = 44), "mostly straight" (n = 42), "bisexual leaning straight" (n = 15), "bisexual" (n = 18), "bisexual leaning lesbian" (n = 13), "mostly lesbian" (n = 26), or "exclusively lesbian" (n = 41). The mean (SD) age of the sample was 24.22 (6.98), and most were Caucasian (78%), followed by Black (6%), Chinese (5%), and other ethnicities. Some participants opted not to self-report their masculinity-femininity, be video-recorded for observer ratings, or have their digits measured (see Procedure). Due to this, data were available for 196 women for self-reports, 191 women for observer ratings, and 182 women for 2D:4D measures, and numbers of participants varied across analyses. A full listing of all descriptive statistics for these 199 participants, including all three measures of masculinity-femininity and both left- and right-hand 2D:4D, can be found in Table 1.

**Raters.** Psychology students participated as raters of masculinity-femininity for course credit. In total we had 48 heterosexual male raters, 21 nonheterosexual male raters, 71 heterosexual female raters, and 29 nonheterosexual female raters. The higher proportion of female raters reflects the fact that in our department, the majority of students are female. Ratings of masculinity-femininity are minimally affected by the raters' sex and sexual orientation [24], and this was also the case in the present research.

### Measures and materials

**Self-reported sexual orientation.** Participants reported their sexual orientation and sexual attraction to men and women on 7-point scales [29]. These scales were highly correlated,

**Table 1. Means, confidence intervals, standard deviations and sample sizes for variables, split by sexual orientation groups.**

| | Left-Hand 2D:4D | Right-Hand 2D:4D | Self-Reported Childhood Masculinity | Self-Reported Adulthood Masculinity | Observer-Rated Adulthood Masculinity |
|---|---|---|---|---|---|
| Heterosexual | .973 [.965, .981] | .975 [.965, .985] | 2.99 [2.63, 3.34] | 2.31 [2.01, 2.61] | 2.78 [2.59, 2.96] |
| (Kinsey 0–1) | (.037, N = 82) | (.044, N = 83) | (1.65, N = 85) | (1.41, N = 85) | (.86, N = 85) |
| Bisexual | .990 [.974, 1.00] | .990 [.977, 1.00] | 3.63 [3.17, 4.10] | 2.73 [2.35, 3.11] | 3.11 [2.86, 3.36] |
| (Kinsey 2–4) | (.047, N = 38) | (.040, N = 38) | (1.81, N = 45) | (1.26, N = 45) | (.81, N = 43) |
| Homosexual | .970 [.961, .979] | .967 [.959, .974] | 3.87 [3.42, 4.31] | 3.41 [2.98, 3.84] | 3.67 [3.34, 4.00] |
| (Kinsey 5–6) | (.035, N = 62) | (.029, N = 62) | (1.80, N = 66) | (1.75, N = 66) | (1.30, N = 63) |

*Note*. Numbers in square brackets represent 95% confidence intervals of the mean. Numbers in round brackets represent standard deviations of the mean and sample sizes. For descriptive purposes, categorizations are based on the Kinsey sexual orientation score, with Kinsey 0–1 considered heterosexual, 2–4 considered bisexual, and 5–6 considered homosexual.

$p < .0001$, $r = .97$, 95% CI [.93, 1.0], and averaged within participants. For this average, a score of 0 represented exclusive heterosexuality, a score of 3 bisexuality with equal attractions, and 6 represented exclusive homosexuality. This average score was treated as continuous variable, simply named "sexual orientation," in all analyses. One exception is the list of descriptive statistics seen in Table 1. For that purpose, we grouped participants into three groups (heterosexual, bisexual, homosexual) based on their original sexual orientation score (before averaging it with attraction).

**Self-reported masculinity-femininity.** Childhood masculinity was assessed using six items from the Childhood Gender Nonconformity Scale [9], and adulthood masculinity was measured using six items from the Continuous Gender Identity Scale [9]. Each scale consists of statements such as "As a child I was perceived as masculine by my peers." for childhood, and "My mannerisms are not very feminine" for adulthood. Responses were given on 7-point scales ranging from 1 (strongly disagree) to 7 (strongly agree), with answers recoded such that higher numbers always represented greater masculinity. Item reliability (Cronbach's alpha) was .89 for the Childhood Gender Nonconformity Scale and .92 for the Continuous Gender Identity Scale.

**2D:4D.** Digit measurements were taken from either high-resolution photographs or scans of participants' hands. For the photographs, participants placed their hands on a flat surface in a supinated (palms facing up) position, with their fingers slightly spread apart, and images were taken from approximately 30 cm above this surface. For the scans, participants placed their hands flat in a pronated (palms facing down) position on the surface of the scanner. In both cases, the palmar surfaces of the hands were visible in the resultant images. Different methods of capturing images (photograph or scanner) did not moderate the relationship between sexual orientation and 2D:4D.

Using these images, digit ratios were measured by two independent raters who were blind to the participants' sex and sexual orientation. Measurements were performed with the vector graphics package Inkscape 0.92, as computer-assisted techniques produce the most reliable measurements [30]. Each rater drew a line as wide as the finger along the proximal skin crease at the base of the finger, between the metacarpal and proximal phalanx. A second line was drawn downwards from the tip of the finger, where it automatically snapped to the center of the base line. Raters then zoomed in on the tip of the finger for fine adjustments, to ensure that this line matched the tip as closely as possible. Measurements for each digit were averaged between raters, as inter-rater reliability (Cronbach's alpha) exceeded .99 for each digit. For each hand, 2D:4D was calculated by dividing the averaged length of the index finger by the averaged length of the ring finger.

## Statistical analysis

All analyses were performed in JMP 14.1.0. We did not exclude any outliers from analyses.

## Procedure

**Participant session.** The University of Essex's Ethics Committee approved this study (GR1303). After providing written informed consent, participants completed a survey on their demographics, sexual orientation, and masculinity-femininity, and had photographs or scans of their hands taken. They were then seated in a chair in front of a white wall and had their entire body video-recorded for 5–10 minutes to capture their gestures and movements. Participants answered questions about the weather, their interests, and their childhood, and were not interrupted while answering. Analyses were based on their answer to a neutral question: "How would you describe the weather at this time of year?" A session took approximately 30 minutes and participants were compensated for their time.

**Editing of participant videos.** The first complete sentence participants spoke in response to the neutral question was extracted using Shotcut. Created clips generally lasted between 10 and 20 seconds. If responses were less than 6 seconds, we took a combination of their first and second sentence. Raters can reliably judge behaviors related to sexual orientation from brief video clips such as these [24, 31].

**Ratings of masculinity-femininity.** Raters, who were blind to the participants' sexual orientation, viewed the edited video clips of target participants. They were instructed to indicate their impression of each woman's appearance and demeanor, in comparison with other women of the same age. For example, they were told to "rate whether this woman appeared or behaved in a more feminine or masculine way". Ratings were completed on 7-point scales, where a score of 1 was "more feminine", 4 "average," and 7 "more masculine." These observer ratings were highly reliable within each rater group (heterosexual and non-heterosexual men and women) and across all raters (all Cronbach's α's > .95). Evaluations were therefore averaged across all raters, producing an average observer-rated masculinity-femininity score for each video-recorded target participant.

# Results

## Hypothesis 1

We hypothesized that homosexual women would be more masculine than heterosexual women. Our measures of masculinity-femininity were self-reports from childhood and adulthood, and observer ratings of adulthood behavior. For each of these measures, we regressed women's masculinity scores onto their sexual orientation, with sexual orientation treated as a continuous variable in all analyses. We originally tested for both linear and curvilinear effects, to account, for example, for the possibility that bisexual women are closer to homosexual than heterosexual women in their masculinity. However, no such patterns were detected, and we focused exclusively on reporting linear effects.

Homosexual women were significantly more masculine than heterosexual women in their self-reports of both childhood, $p < .001$, $\beta = .24$, 95% CI [.11, .38] (Fig 1A), and adulthood, $p < .001$, $\beta = .31$ [.18, .45] (Fig 1B), and in observer ratings of their behavior in adulthood, $p < .001$, $\beta = .38$ [.25, .51] (Fig 1C).

## Hypothesis 2

We hypothesized that homosexual women would have more male-typical (lower) 2D:4D than heterosexual women in both their left and right hand. We regressed women's left hand and

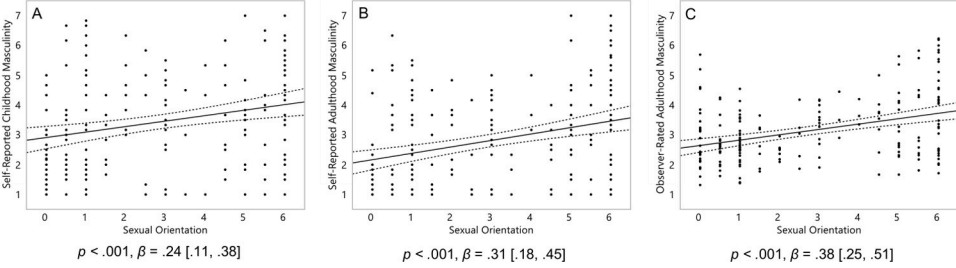

**Fig 1.** Masculinity of 196 women (self-report from childhood, A; and adulthood, B) and 191 women (observer-ratings, C). On the Y axis, masculinity scores, with higher scores representing a greater degree of masculinity. On the X axis, 0 represents exclusive heterosexuality, 3 represents bisexuality, and 6 represents exclusive homosexuality. Triple lines represent regression coefficients with their 95% confidence intervals. Dots represent participants' scores. Statistics represent linear effects.

right hand 2D:4D onto their sexual orientation. Although in the predicted directions, homosexual women did not have a significantly lower 2D:4D than heterosexual women in either their left hand, $p = .26$, $\beta = -.08$ [-.23, .06], or their right hand, $p = .67$, $\beta = -.03$ [-.18, .11] (Fig 2A and 2B). Thus, our hypothesis that homosexual women would show signs of exposure to elevated levels of prenatal androgens was not supported.

## Hypothesis 3

We hypothesized that the relationship between women's sexual orientation and their degree of masculinity was mediated by their male-typical 2D:4D. Although 2D:4D did not significantly link to sexual orientation, we still conducted this analysis as it was planned in advance. We computed multiple regression analyses. We predicted one of the three masculinity-femininity variables by sexual orientation in Step 1, and by sexual orientation plus left-hand 2D:4D as a mediator in Step 2. We chose to focus on left-hand 2D:4D as it was closer to significance than right-hand 2D:4D (Fig 3A). However, we did conduct similar analyses with right-hand 2D:4D, and as expected, it did not influence any effects of sexual orientation on masculinity.

Table 2 summarizes the results of the analyses for all three masculinity variables. The effect of sexual orientation on masculinity remained similar in magnitude and levels of significance

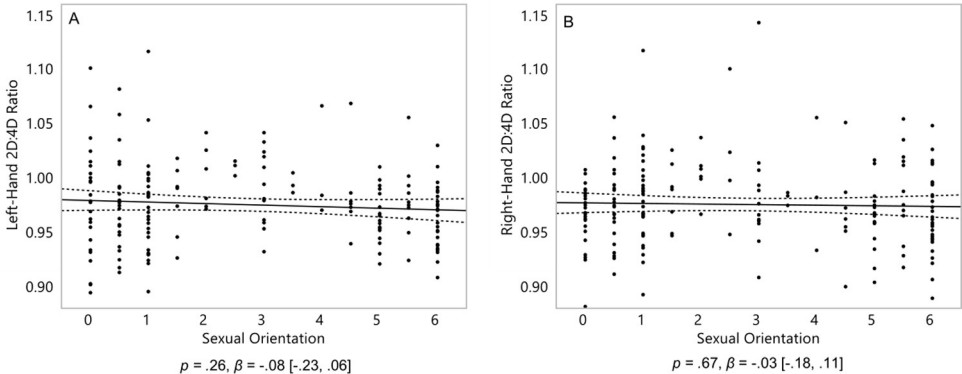

**Fig 2.** 2D:4D of 182 women (left-hand 2D:4D, A; and right-hand 2D:4D, B). On the Y axis, 2D:4D is the length of the index finger divided by the length of the ring finger, with lower scores representing a more male-typical 2D:4D. On the X axis, 0 represents exclusive heterosexuality, 3 represents bisexuality, and 6 represents exclusive homosexuality. Triple lines represent regression coefficients with their 95% confidence intervals. Dots represent participants' scores. Statistics represent linear effects.

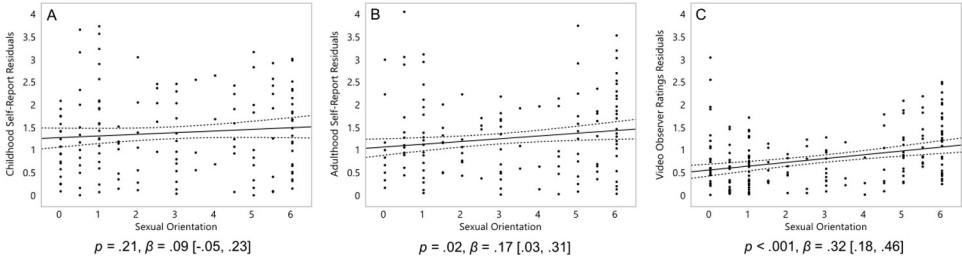

**Fig 3. Masculinity of 196 women.** Absolute residuals derived from the effects displayed in Fig 1 for masculinity of 196 women (self-report from childhood, A; and adulthood, B) and 191 women (observer-ratings, C). On the Y axis, residuals for masculinity, with higher scores representing a greater degree of variance from the main effect. On the X axis, 0 represents exclusive heterosexuality, 3 represents bisexuality, and 6 represents exclusive homosexuality. Triple lines represent regression coefficients with their 95% confidence intervals. Dots represent participants' residuals. Statistics represent linear effects.

before and after the inclusion of left-hand 2D:4D as a second predictor. Hence, thus far, there was no evidence that 2D:4D mediated the relationship between sexual orientation and masculinity. Yet, systematic mediation analyses were still necessary to confirm this. We therefore computed three mediation analyses (one for each measure of masculinity-femininity) on the basis of 10,000 bootstrapped samples [32]. Left-hand 2D:4D did not significantly mediate the relationship between sexual orientation and masculinity, as the confidence intervals of the mediation effects included zero. This was true for self-recalled childhood masculinity, $\beta = -.005$ [-.02, .01], self-reported adulthood masculinity, $\beta = .004$ [-.001, .03], and observer-rated adulthood masculinity, $\beta = .002$ [-.01, .02].

## Hypothesis 4

We hypothesized that homosexual women would be more variable than heterosexual women in both their masculinity and 2D:4D. To test for their increased variability, we first calculated the residuals for the main effect of sexual orientation on each of the three measures of masculinity-femininity depicted in Hypothesis 1 (Fig 1A–1C), and each of the two main effects of sexual orientation on 2D:4D depicted in Hypothesis 2 (Fig 2A and 2B). We then computed the

**Table 2. Multiple regression analyses for sexual orientation and left-hand 2D:4D predicting self-reported childhood and adulthood masculinity (Step 1 N = 196, Step 2 N = 181) and observer-rated adulthood masculinity (Step 1 N = 191, Step 2 N = 180).**

| Step 1 | Self-Reported Childhood Masculinity | Self-Reported Adulthood Masculinity | Observer-Rated Adulthood Masculinity |
|---|---|---|---|
| **Variables** | $\beta$ | $\beta$ | $\beta$ |
| Sexual Orientation (SO)[1] | .24 [.11, .38]** | .31 [.18, .45]** | .38 [.25, .51]** |
| **Step 2** | **Self-Reported Childhood Masculinity** | **Self-Reported Adulthood Masculinity** | **Observer-Rated Adulthood Masculinity** |
| **Variables** | $\beta$ | $\beta$ | $\beta$ |
| Sexual Orientation (SO)[1] | .25 [.10, .39]** | .29 [.15, .43]** | .36 [.22, .50]** |
| Left-Hand 2D:4D[2] | .07 [-.08, .21] | -.05 [-.19, .09] | -.03 [-.16, .11] |

*Note. $R^2$'s for the three models are .06, .10 and .14 in Step 1; .06, .09 and .12 in Step 2, respectively. Numbers in brackets represent 95% confidence intervals of the standardized regression coefficient, $\beta$.*

[1]Higher scores indicate a more homosexual orientation.

[2]Lower scores indicate a more male-typical 2D:4D.

[†] $p < .1$

[*] $p < .05$

[**] $p < .01$.

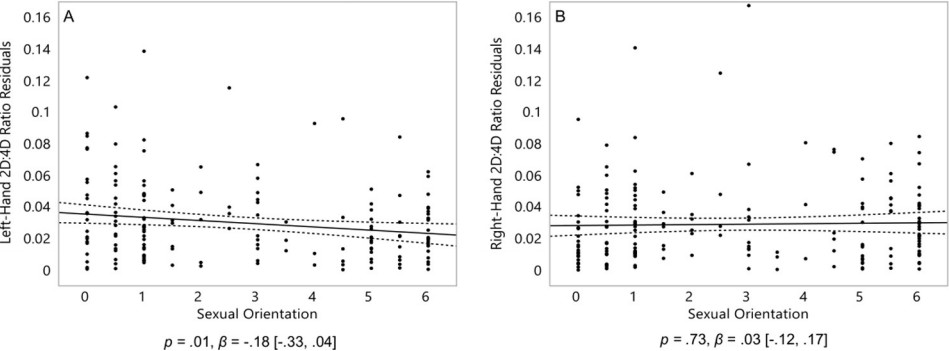

**Fig 4. 2D:4D of 182 women.** Absolute residuals derived from the effects displayed in Fig 2 for 2D:4D (left-hand 2D:4D, A; and right-hand 2D:4D, B). On the Y axis, residuals for 2D:4D, with higher scores representing a greater degree of variance from the main effect. On the X axis, 0 represents exclusive heterosexuality, 3 represents bisexuality, and 6 represents exclusive homosexuality. Triple lines represent regression coefficients with their 95% confidence intervals. Dots represent participants' residuals. Statistics represent linear effects.

absolute values of these residuals. Finally, we performed Levene's tests for unequal variances to establish whether the degree of variance (magnitude of absolute residuals) in these effects were stronger in homosexual women than heterosexual women.

Homosexual women were not significantly more variable than heterosexual women in their self-reported childhood masculinity, $p = .21$, $\beta = .09$ [-.05, .23] (Fig 3A), but were significantly more variable than heterosexual women in both their self-reports of adulthood, $p = .02$, $\beta = .17$ [.03, .31] (Fig 3B), and observer-ratings of their videos from adulthood, $p < .001$, $\beta = .32$ [.18, .46] (Fig 3C).

Contrary to our hypothesis, homosexual women were significantly *less* variable in their left-hand 2D:4D than heterosexual women, $p = .01$, $\beta = -.18$ [-.33, .04] (Fig 4A). They were not significantly more (or less) variable in their right-hand 2D:4D than heterosexual women, $p = .73$, $\beta = .03$ [-.12, .17] (Fig 4B).

Thus, even though we did not find any evidence for homosexual women's increased variability in the marker of prenatal androgen exposure, they were more variable in two out of three measures of masculinity-femininity. This increased variance in masculinity-femininity within homosexual women still pointed to the possibility that there are different types of homosexual women, who may be differentiated by the measure of androgen exposure. We examined this possibility in our next analyses.

## Hypothesis 5

We hypothesized that homosexual women with the most male-typical 2D:4D show the greatest degree of masculinity, as compared to heterosexual women, and other homosexual women with less male-typical 2D:4D. We calculated three multiple regression analyses, predicting one of our three measures of masculinity-femininity. In each analysis, independent variables were sexual orientation, left-hand 2D:4D, and their interaction. If variation in 2D:4D explains why homosexual women are more variable in masculinity-femininity than heterosexual women, then this interaction between sexual orientation and left-hand 2D:4D will be significant.

The results of the regression analyses are summarized in Table 3. For all three measures of masculinity-femininity, sexual orientation was the only significant predictor of masculinity-femininity. Neither the effect of left-hand 2D:4D, nor the interaction of sexual orientation with 2D:4D were significant in any of the analyses, and the standardized regression coefficients of these effects were weak in magnitude.

**Table 3. Multiple regression analyses for sexual orientation and left-hand 2D:4D predicting self-reported childhood and adulthood masculinity (N = 181) and observer-rated adulthood masculinity (N = 180).**

| Step 1 | Self-Reported Childhood Masculinity | Self-Reported Adulthood Masculinity | Observer-Rated Adulthood Masculinity |
|---|---|---|---|
| Variables | β | β | β |
| Sexual Orientation (SO)[1] | .25 [.10, .39]** | .28 [.14, .43]** | .36 [.22, .50]** |
| Left-Hand 2D:4D[2] | .07 [-.08, .22] | -.07 [-.22, .08] | -.03 [-.18, .12] |
| SO x 2D:4D | .01 [-.14, .16] | -.05 [-.20, .10] | -.01 [-.16, .13] |

*Note.* $R^2$'s for the three models are .06, .09 and .13, respectively. Numbers in brackets represent 95% confidence intervals of the standardized regression coefficient, β.

[1]Higher scores indicate a more homosexual orientation.

[2]Lower scores indicate a more male-typical 2D:4D.

† $p < .1$

* $p < .05$

** $p < .01$.

## Discussion

Present findings confirmed that homosexual women were more masculine than heterosexual women, on average. Furthermore, homosexual women were more variable in their masculinity in two out of three measures. However, contrary to our hypothesis, they were significantly less variable in their left-hand (but not right-hand) 2D:4D than heterosexual women, and we do not have any reasonable explanation for this. Furthermore, homosexual women did not have more male-typical digit ratios, nor did their 2D:4D mediate or moderate the relationship between sexual orientation and their degree of masculinity. The finding that homosexual women are more masculine, in general, but also more variable in their masculinity-femininity than heterosexual women has been previously reported [2]. In combination, these findings point to the possible existence of different types of homosexual women, at least with respect to their masculinity-femininity. Hence, it seemed conceivable the most masculine homosexual women, especially, would show signs of increased androgen exposure in the form of more male-typical 2D:4D. However, this was not the case in the present sample. In general, 2D:4D was not significantly more masculine in homosexual women than heterosexual women, even though the effect was in the predicted direction. This is noteworthy, as such a pattern was previously confirmed in a meta-analysis [16]. Perhaps our measure, 2D:4D, was not sensitive enough to robustly indicate prenatal androgen exposure. Yet, we consider this unlikely, as we have previously confirmed a sexual orientation difference in 2D:4D in a much smaller sample of women using the identical methodology [20]. Furthermore, as mentioned previously, computer-assisted measurement techniques, such as those employed in the current study, are highly reliable [30] and this was also the case in the present study. Finally, even though men were not the focus of the present research, we had simultaneously gathered 2D:4D data from male participants for a different project. As predicted, these men had significantly lower (more male-typical) 2D:4D than women in both the left hand, $d = .33$ [.33, .34], $p = .004$, and right hand, $d = .33$ [.33, .33], $p = .008$. Thus, it seems less likely that present null findings are a result of measurement issues. Maybe, in the present study, homosexual women simply did not have more male-typical digit ratios than heterosexual women.

Perhaps the present research should have used a self-report measure of "butch" and "femme" identities, rather than degrees of masculinity-femininity, in order to elicit the hypothesized effects, as such an approach succeeded in previous work [26]. Yet, because we reasonably assumed that women who self-identify as "butch" would be more masculine compared to those who identifies as "femme" [26], we expected that the present measures of masculinity-femininity would reveal predicted effects, if they were indeed present.

It is further possible that 2D:4D is not a sensitive enough measure to significantly explain the relationship between sexual orientation and masculinity-femininity. Indeed, there is ongoing debate about the utility and strength of 2D:4D as a measure of androgen exposure [21, 22]. Perhaps, future research may have more success using other biomarkers of prenatal androgen exposure, in addition to 2D:4D. Another indirect measure of prenatal androgenization is the finger ridge count. This measure has previously indicated that homosexual female monozygotic twins may have been more masculinized than their heterosexual co-twins [33], and the measure itself correlates with 2D:4D [34]. Other potential measures may include otoacoustic emissions [35] or anogenital distance [36]. These measures show promise for measuring prenatal androgen exposure, with otoacoustic emissions producing a difference between heterosexual and homosexual women which had a larger effect size than the difference in 2D:4D found in a meta-analysis, $d = .23$ (left) and .29 (right) for 2D:4D, and $d = .37$ for otoacoustic emissions [16, 35].

A related limitation concerns statistical power in the present study. The sample size we chose was based on the weakest estimated effect: The relationship between left-hand 2D:4D and sexual orientation ($r = .30$). This estimate was taken from a previous research project which found a significant effect using the exact same measurement procedures conducted by the same researchers in the same lab [20]. However, this previous project used identical twins as participants. Although we treated these twins as unrelated (i.e. unpaired) in our power calculations for the present study, it was perhaps naïve of us to assume that the effect previously found in twins would be equally strong in unrelated participants. In the present research, the strongest effect was $r$ (or $\beta$) = -.08 in the left hand. With this effect, post-hoc power analyses suggested that we would have needed a minimum sample size of 1634 women of different sexual orientations for it to become significant. If our a-priori sample size estimate had returned such a large number, we would have considered it an unreasonable goal for a lab-based study like ours.

Perhaps, also, we should have considered in advance the relationships between sexual orientation and 2D:4D calculated by the meta-analysis of Grimbos et al. (2010). In this respect, it is worth noting that for the present study, once effect sizes were converted into Hedge's g (the effect size used in the meta-analysis), our estimates of the relationship between sexual orientation and 2D:4D were -.16 for the left hand, and -.06 for the right hand. These estimates fall within the 95% confidence intervals (but closer to zero) of the unadjusted meta-analytic estimates; which were (scaled in the same direction as present effects), $g = -.23$ [-.51, -.06] for the left hand, and $g = -.29$ [-.43, -.04] for the right hand. Our estimates were also close to the publication bias-corrected estimates given in the same meta-analysis, which were -.07 for the left hand and -.13 for the right hand.

Additionally, as mentioned in the method section, it is possible that the present study was not sufficiently powerful for 2D:4D to mediate or moderate the relationship between sexual orientation and masculinity-femininity. However, the effect sizes for the computed mediations and moderations (e.g., Table 3) were so small in magnitude that the most parsimonious assessment from the present data is that it seems unlikely to detect such patterns even in much larger samples.

A final point concerns bisexual women, who were intermediate between heterosexual and homosexual women in both measures and variability of their masculinity-femininity. That is, our analyses indicated that the relationships of sexual orientation with masculinity-femininity were explained by simple linear effects, whereas we found no evidence for curvilinear effects that would, for instance, suggest that bisexual women are closer to homosexual women than heterosexual women in their masculinity-femininity. Although it seems sensible that bisexual women are between these two groups with respect to their masculinity-femininity, there are no strong hypotheses regarding what factors would cause this outcome [37].

In sum, the present study did not find evidence of a link between masculinity in homosexual women and exposure to androgens in the prenatal period, as reflected in finger length ratios. In fact, in the present study, homosexual women showed no clear signs of elevated prenatal androgen exposure, as compared to heterosexual women. Thus, our hypothesis that homosexual women's male-typed traits were influenced by early hormonal influences remains unconfirmed.

## Author Contributions

**Conceptualization:** Luke Holmes, Gerulf Rieger.

**Data curation:** Luke Holmes.

**Formal analysis:** Luke Holmes, Gerulf Rieger.

**Funding acquisition:** Gerulf Rieger.

**Investigation:** Luke Holmes, Tuesday M. Watts-Overall, Erlend Slettevold, Dragos C. Gruia, Gerulf Rieger.

**Methodology:** Luke Holmes, Gerulf Rieger.

**Project administration:** Luke Holmes, Gerulf Rieger.

**Supervision:** Gerulf Rieger.

**Validation:** Luke Holmes.

**Visualization:** Luke Holmes.

**Writing – original draft:** Luke Holmes.

**Writing – review & editing:** Luke Holmes, Tuesday M. Watts-Overall, Erlend Slettevold, Dragos C. Gruia, Gerulf Rieger.

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
