## [Decision Letter · Decision Letter 0]

7 Oct 2021

PONE-D-21-00755

Digit ratios and masculinity-femininity of women

PLOS ONE

Dear Dr. Holmes,

Thank you for submitting your manuscript to PLOS ONE. After careful consideration, we feel that it has merit but does not fully meet PLOS ONE’s publication criteria as it currently stands. Therefore, we invite you to submit a revised version of the manuscript that addresses the points raised during the review process.

As you see, both reviewers are pretty happy with your manuscript. However, they require more or less technical improvements. Specifically, you should clear up the Kinsey scores and add a table summary statistics for 2d:4d. It would also help if you mentioned other biomarkers for detecting prenatal hormone exposure. Please also consider J Tortorice's thesis, mentioned by Reviewer 2.

If you were willing to revise the manuscript as suggested, it is likely I may accept it without sending it to the reviewers again.

We look forward to receiving your revised manuscript.

Kind regards,

Ludek Bartos

Academic Editor

PLOS ONE

Journal Requirements:

3. Please modify the title to ensure that it is meeting PLOS’ guidelines (https://journals.plos.org/plosone/s/submission-guidelines#loc-title). In particular, the title should be "specific, descriptive, concise, and comprehensible to readers outside the field" and in this case  it is not informative and specific about your study's scope and methodology.

Reviewers' comments:

Reviewer's Responses to Questions

**Comments to the Author**

1. Is the manuscript technically sound, and do the data support the conclusions?

Reviewer #1: Yes

Reviewer #2: Yes

2. Has the statistical analysis been performed appropriately and rigorously? 

Reviewer #1: Yes

Reviewer #2: Yes

3. Have the authors made all data underlying the findings in their manuscript fully available?

Reviewer #1: Yes

Reviewer #2: Yes

4. Is the manuscript presented in an intelligible fashion and written in standard English?

Reviewer #1: Yes

Reviewer #2: Yes

5. Review Comments to the Author

Reviewer #1: This paper is designed to test hypotheses that homosexual women are more masculine than hetero- sexual women. They use finger length ratio, 2dL4d to test the hypothesis that prenatal hormones are responsible for masculinization. I found the paper well organized, well written, with the hypotheses clearly stated and explicitly tested in the results. It examines the relationship between sexual orientation in females and masculinity-femininity scores, and to 2d:4d. The results support the hypothesis that homosexual women have higher masculinity scores than heterosexual women. It contributes to the relatively sparse literature examining the relationship between sexual orientation and 2d:4d. That their results did not support a relationship adds to the uncertainty surrounding this question. Overall, I found this research to have been carefully thought out, the methods stringent, and the results placed into context. I therefore think it is worthy of publication. I have some comments that I think would improve the paper, as follows:

1. P. 11, lines 245-246, authors define Kinsey scores as 0-1=heterosexual, 2-4=bisexual, and 5-6=homosexual. But in Figures 1, 2 and 3 there appear to be fractions at 1.5, 2.5, etc. Does 1.5 go with heterosexual or bisexual? For the regression analysis it does not matter because sexual orientation is a continuous variable. But the scale should be clarified so readers can understand the values on the X axis.

2. Figure 2 there appear to be outlying values, especially for the right hand. The most extreme value, associated with score 3, appears to approach 1.15. This is probably more than 3 standard deviations above the means of the groups. Authors should make sure the extreme values are not due to measurement error or some other artifact. If they are accurate values, authors may consider removing them.

3. Following up on point 2, I think it would be useful if the authors would present a table of summary statistics for 2d:4d, perhaps by their heterosexual, bisexual and homosexual scores. That would enable readers to relate them to other studies.

4. p. 19, lines 385-391. Authors suggest other biomarkers for detecting prenatal hormone exposure. They could also consider the potential of finger ridge-counts in this role. A difference between monozygotic twins discordant for sexual orientation has been identified. See Hall, L. S. (2000). Dermatoglyphic analysis of total finger ridge count in female monozygotic twins discordant for sexual orientation. Journal of Sex Research, 37(4), 315-320. Ridge-counts have also been shown to correlated with 2d:4d, to some degree involving digits 2 and 4. See Jantz, R. L. (2021). Finger ridge-counts correlate with the second to fourth digit ratio (2d:4d). American Journal of Human Biology, e23625.

Reviewer #2: The authors examine a sample of adult women and find correlations between self-reported and observer estimated masculinity-feminity, and found that homosexual women were more masculine than straight women by both measures. They did not see a relationship between sexual orientation and digit ratios, nor between digit ratios and either measure of masculinity. I presume the bulk of the women were recruited from somewhere in the UK, given the authors affiliations, but they should specify this in the methods section. The results match some previous reports and not others (including some results from their own lab), but the authors do an admirable job of critiquing themselves and offering possible explanations for the discrepancies. As far as I can tell, all the measures were done well, the statistical analyses are appropriate, and the writing is clear and direct. If the authors can secure J Tortorice's thesis on butch-femme differences on digit ratio, they might cite that, but if they cannot get it, then of course they should not cite it.

6. PLOS authors have the option to publish the peer review history of their article (what does this mean?). If published, this will include your full peer review and any attached files.

Reviewer #1: **Yes: **Richard Jantz

Reviewer #2: No

---

## [Author Response · Author response to Decision Letter 0]

12 Oct 2021

The following is the same text as in the Response to Reviewers file attached to the revision application. Please use whichever is more convenient for you. 

Dear Dr. Bartos,

Thank you for the opportunity to revise our manuscript PONE-D-21-00755, (previously) entitled “Digit ratios and masculinity-femininity of women”, for PLOS ONE. In the following, we detail how we addressed the comments made by yourself and the reviewers. In order to avoid confusion, we have re-numbered some of the lists, but all points raised are still listed.

Sincerely,

The Authors

Editor:

E.1. Thank you for submitting your manuscript to PLOS ONE. After careful consideration, we feel that it has merit but does not fully meet PLOS ONE’s publication criteria as it currently stands. Therefore, we invite you to submit a revised version of the manuscript that addresses the points raised during the review process.

We are grateful for this positive appraisal of our paper, and we hope that the following addresses all of your concerns. 

E.2. As you see, both reviewers are pretty happy with your manuscript. However, they require more or less technical improvements. Specifically, you should clear up the Kinsey scores and add a table summary statistics for 2d:4d. It would also help if you mentioned other biomarkers for detecting prenatal hormone exposure. Please also consider J Tortorice's thesis, mentioned by Reviewer 2.

If you were willing to revise the manuscript as suggested, it is likely I may accept it without sending it to the reviewers again.

We thank the editor for this positive appraisal of the review. In the following, we outline how we have addressed all of these concerns: In R1.2 and R1.4 we detail how we removed the confusing section on categorization via Kinsey scores, and replaced this with an easy-to-read table which lists all of the necessary descriptive statistics for the benefit of future research (Table 1). We agree that the papers on another biomarker suggested in R1.5. are appropriate for our research, and have added them to the discussion. Similarly, we managed to obtain a copy of J Tortorice’s thesis, mentioned in R2.4., and have similarly added this to the appropriate section in the introduction. 

E.3. Please ensure that your manuscript meets PLOS ONE's style requirements, including those for file naming. The PLOS ONE style templates can be found at 

Our manuscript now adheres fully to the PLOS ONE style guide.

E.4. Please review your reference list to ensure that it is complete and correct. If you have cited papers that have been retracted, please include the rationale for doing so in the manuscript text, or remove these references and replace them with relevant current references. Any changes to the reference list should be mentioned in the rebuttal letter that accompanies your revised manuscript. If you need to cite a retracted article, indicate the article’s retracted status in the References list and also include a citation and full reference for the retraction notice.

We have checked through the reference list, and it is complete and correct. The one change made is that we now cite two new papers suggested in R1.5. and the thesis of J Tortorice, suggested in R2.4. The total number of references has therefore risen from 34 to 37.

E.5. Please modify the title to ensure that it is meeting PLOS’ guidelines (https://journals.plos.org/plosone/s/submission-guidelines#loc-title). In particular, the title should be "specific, descriptive, concise, and comprehensible to readers outside the field" and in this case it is not informative and specific about your study's scope and methodology.

We have considered the example titles, and have decided to rename the paper “The relationship between finger length ratio, masculinity, and sexual orientation in women: A correlational study”. We hope that this covers the employed variables and methodology of the paper in sufficient detail. The website states also that we should include a shortened title, but we are not sure where to insert it because the style guide does not include it. Based on the examples given, we hope that the original title (“Digit ratios and masculinity-femininity of women”) is suitable for this purpose if needed. 

Reviewer #1: 

R1.1. This paper is designed to test hypotheses that homosexual women are more masculine than hetero- sexual women. They use finger length ratio, 2dL4d to test the hypothesis that prenatal hormones are responsible for masculinization. I found the paper well organized, well written, with the hypotheses clearly stated and explicitly tested in the results. It examines the relationship between sexual orientation in females and masculinity-femininity scores, and to 2d:4d. The results support the hypothesis that homosexual women have higher masculinity scores than heterosexual women. It contributes to the relatively sparse literature examining the relationship between sexual orientation and 2d:4d. That their results did not support a relationship adds to the uncertainty surrounding this question. Overall, I found this research to have been carefully thought out, the methods stringent, and the results placed into context. I therefore think it is worthy of publication. I have some comments that I think would improve the paper, as follows:

We thank the reviewer for this highly positive appraisal of our research. We hope that the following changes address your concerns. 

R1.2. P. 11, lines 245-246, authors define Kinsey scores as 0-1=heterosexual, 2-4=bisexual, and 5-6=homosexual. But in Figures 1, 2 and 3 there appear to be fractions at 1.5, 2.5, etc. Does 1.5 go with heterosexual or bisexual? For the regression analysis it does not matter because sexual orientation is a continuous variable. But the scale should be clarified so readers can understand the values on the X axis.

We have now deleted the references to Kinsey score in this section (line 262), as it immediately follows a sentence stating that we use sexual orientation as a continuous variable throughout all analyses, and thus would cause needless confusion to the reader. Instead, we now explain in the Self-reported sexual orientation section of the Method (line 190) that we use the averaged (Sexual Orientation + Attraction) sexual orientation score for all analyses. We averaged them as they were highly correlated, and their average will therefore give the most reliable scores. However, for descriptive purposes only, when categorizing participants (which only applies in Table 1), we instead use the Kinsey score alone to avoid participants falling “in between” categories in a confusing manner. We mention this both in the Method (line 191), and again in Table 1 itself. 

R1.3. Figure 2 there appear to be outlying values, especially for the right hand. The most extreme value, associated with score 3, appears to approach 1.15. This is probably more than 3 standard deviations above the means of the groups. Authors should make sure the extreme values are not due to measurement error or some other artifact. If they are accurate values, authors may consider removing them.

This is true – there were outlying 2D:4D values, and these were especially prominent for one bisexual participant. Ultimately, we concluded that they should not be removed for several reasons: Firstly, we have checked extensively that these values were not down to some form of measurement error, and we are satisfied that these participants simply have especially short ring fingers on their right hands. Thus, for purely theoretical reasons, we are hesitant to remove them. As this finger was the source of the high values, it could simply indicate abnormally low androgenisation in these participants. Secondly, the removal or inclusion of these participants did not create a statistically significant relationship between sexual orientation and 2D:4D. For example, for the right hand, this relationship was not significant with three main outliers included, p = .67, β = -.03 [-.18, .11], and it remained non-significant with them excluded, p = .80, β = -.02 [-.17, .13]. Furthermore, inclusion or exclusion did not impact the mediation or moderation analyses. Taken in combination, we decided based on these factors that the study was more robust with these participants included and depicted on the graphs. 

R1.4. Following up on point 2, I think it would be useful if the authors would present a table of summary statistics for 2d:4d, perhaps by their heterosexual, bisexual and homosexual scores. That would enable readers to relate them to other studies.

We have now included a new table (Table 1) which lists the summary statistics for left- and right-hand 2D:4D, as well as all three measures of masculinity-femininity, separated by Kinsey score. We now direct readers to this table in the Participants section (line 174). We explain the reasoning for using Kinsey score as a categorising variable in R1.2., and we explain in the paper that it is used in this manner in the Sexual Orientation section of the Method (line 191), as well as in the caption of Table 1. 

R1.5. p. 19, lines 385-391. Authors suggest other biomarkers for detecting prenatal hormone exposure. They could also consider the potential of finger ridge-counts in this role. A difference between monozygotic twins discordant for sexual orientation has been identified. See Hall, L. S. (2000). Dermatoglyphic analysis of total finger ridge count in female monozygotic twins discordant for sexual orientation. Journal of Sex Research, 37(4), 315-320. Ridge-counts have also been shown to correlated with 2d:4d, to some degree involving digits 2 and 4. See Jantz, R. L. (2021). Finger ridge-counts correlate with the second to fourth digit ratio (2d:4d). American Journal of Human Biology, e23625.

We thank the reviewer for these relevant references. Ridge count is now listed among the other potential biomarkers (line 405), and these two papers are cited there. 

Reviewer #2: 

R2.1. The authors examine a sample of adult women and find correlations between self-reported and observer estimated masculinity-femininity, and found that homosexual women were more masculine than straight women by both measures. They did not see a relationship between sexual orientation and digit ratios, nor between digit ratios and either measure of masculinity. 

We appreciate the reviewer’s detailed examination of our work. 

R2.2. I presume the bulk of the women were recruited from somewhere in the UK, given the authors affiliations, but they should specify this in the methods section. 

We now specify that the women in the study were recruited from Colchester and London in the United Kingdom (Line 164).

R2.3. The results match some previous reports and not others (including some results from their own lab), but the authors do an admirable job of critiquing themselves and offering possible explanations for the discrepancies. As far as I can tell, all the measures were done well, the statistical analyses are appropriate, and the writing is clear and direct.

We thank the reviewer for this positive appraisal of our paper. 

R2.4. If the authors can secure J Tortorice's thesis on butch-femme differences on digit ratio, they might cite that, but if they cannot get it, then of course they should not cite it.

We have managed to procure a digital copy of this thesis, which is relevant to our work. We now include this study, along with a brief description of its methodology and findings, in the appropriate section of the introduction (Page 106).

---

## [Editor Report · Decision Letter 1]

25 Oct 2021

The relationship between finger length ratio, masculinity, and sexual orientation in women: A correlational study

PONE-D-21-00755R1

Dear Dr. Holmes,

We’re pleased to inform you that your manuscript has been judged scientifically suitable for publication and will be formally accepted for publication once it meets all outstanding technical requirements.

Kind regards,

Ludek Bartos

Academic Editor

PLOS ONE
---

## [Editor Report · Acceptance letter]

28 Oct 2021

PONE-D-21-00755R1 

The relationship between finger length ratio, masculinity, and sexual orientation in women: A correlational study 

Dear Dr. Holmes:

I'm pleased to inform you that your manuscript has been deemed suitable for publication in PLOS ONE. Congratulations! Your manuscript is now with our production department. 

Kind regards, 

on behalf of

Dr. Ludek Bartos 

Academic Editor

PLOS ONE